# Optimization of Sperm Cryopreservation Protocol for Mediterranean Brown Trout: A Comparative Study of Non-Permeating Cryoprotectants and Thawing Rates In Vitro and In Vivo

**DOI:** 10.3390/ani9060304

**Published:** 2019-05-31

**Authors:** Giusy Rusco, Michele Di Iorio, Pier Paolo Gibertoni, Stefano Esposito, Maurizio Penserini, Alessandra Roncarati, Silvia Cerolini, Nicolaia Iaffaldano

**Affiliations:** 1Department of Agricultural, Environmental and Food Sciences, University of Molise, 86100 Campobasso CB, Italy; g.rusco@studenti.unimol.it (G.R.); michele.diiorio@unimol.it (M.D.I.); 2Mediterranean Trout Research Group—Centro di ricerche “I Giardini dell’Acqua”, 42037 Collagna RE, Italy; gibertoni@igiardinidellacqua.com (P.P.G.); dott.stefanoesposito@gmail.com (S.E.); mauriziopenserini.bio@gmail.com (M.P.); 3School of Biosciences and Veterinary Medicine, University of Camerino, 62032 Camerino MC, Italy; alessandra.roncarati@unicam.it; 4Department of Veterinary Medicine, University of Milan, 20122 Milano MI, Italy; silvia.cerolini@unimi.it

**Keywords:** *Salmo cettii*, sperm freezing, egg yolk, conservation biology, sperm cryobank, fertilization rate

## Abstract

**Simple Summary:**

Cryobanking is an important tool to preserve the genetic resources of fish species. Semen cryopreservation has been extensively used in conservation programs for endangered species. Here, we aimed to find an effective cryopreservation protocol for the autochthonous Mediterranean brown trout inhabiting the Biferno river (south Italy), in order to create a sperm cryobank. Low-density lipoproteins and sucrose were tested as non-permeating cryoprotectants (NP-CPAs) to replace the egg yolk. Moreover, the thawing rate (10 °C for 30 s vs. 30 °C for 10 s) was also studied. From results obtained in vitro and in vivo, egg yolk emerged as the best NP-CPA and the lower thawing rate recorded better post-thaw semen quality in vitro and higher fertilization and hatching rates in vivo. These findings are important because they will contribute to the creation of a sperm cryobank for Molise’s native trout, which is a milestone of our European project (Life Nat.Sal.Mo).

**Abstract:**

The aim of our study was to test the effects of different non-permeating cryoprotectants (NP-CPAs), namely low-density lipoproteins (LDLs), sucrose, and egg yolk, and thawing rates on the post-thaw semen quality and fertilizing ability of the native Mediterranean brown trout. Pooled semen samples were diluted 1:3 (v:v) with 2.5%, 5%, 10%, or 15% LDL; 0.05, 0.1, or 0.3 M sucrose; or 10% egg yolk. At the moment of analysis, semen was thawed at 30 °C/10 s or 10 °C/30 s. The post-thaw semen quality was evaluated, considering motility, the duration of motility, viability, and DNA integrity. Significantly higher values of motility and viability were obtained using egg yolk/10 °C for 30 s, across all treatments. However, LDL and sucrose concentrations affected sperm cryosurvival, showing the highest post-thaw sperm quality at 5% LDL and 0.1 M sucrose. Based on the in vitro data, egg yolk, 5% LDL, and 0.1 M sucrose thawed at 10 °C or 30 °C were tested for the in vivo trial. The highest fertilization and hatching rates were recorded using egg yolk/10 °C (*p* < 0.05). According to these in vitro and in vivo results, egg yolk emerged as the most suitable NP-CPA and 10 °C/30 s as the best thawing rate for the cryopreservation of this trout sperm, under our experimental conditions.

## 1. Introduction

Sperm cryopreservation is considered a valuable tool for preserving the genetic material of endangered fish species by storage of their gametes in a cryobank [1,2,3]. The semen cryobanks provide the opportunity to preserve representative samples and further reconstruct the original strain, population, or diversity [2,4]. In this regard, we attempted to find an effective semen cryopreservation protocol for the Mediterranean brown trout (*Salmo cettii*), a native species that inhabits the rivers of Molise [5]. Currently, this species is listed in the Italian IUNC Red List as critically endangered [6], due to river pollution, uncontrolled fishing, and hybridization following the introduction of non-native strains that have drastically reduced the number of the native species in recent centuries [7].

In our previous works [5,8], the effects of different freezing rates, types of basic extenders, and penetrating cryoprotectants (P-CPAs) were studied in vitro and in vivo. For our native trout, an adequate amount of satisfactory results had been obtained when the semen was frozen in the presence of a glucose-DMSO extender at 5 cm above the liquid nitrogen surface, for 10 min. However, further factors still need to be tweaked to improve the fertilizing ability of cryopreserved semen, in order to achieve satisfactory in vivo results that are closer to those of fresh semen.

In this regard, the optimization of temperature and thawing rate are very important tools to improve the cryopreservation protocol, because thawing rate is a critical factor in preserving the survival of the spermatozoa [9]. Nevertheless, there are few available data present in the literature regarding the thawing conditions for brown trout semen cryopreservation.

Another important factor is finding the best combination between non-permeating cryoprotectant (NP-CPA) and P-CPA in the freezing extender, which can result in a decrease in sperm damages caused by intra and extracellular ice formation, and also improve the egg fertilization rate [10,11]. Egg yolk has been used as an NP-CPA in our previous studies [5,8]. However, in recent years, there have been increasing demands to replace whole egg yolks in semen extenders for two main reasons; one being the presence of substances in the egg yolk that could inhibit the sperm respiration [12,13,14]; the second being the sanitary and practical disadvantages that could occur from its use [15]. In this regard, both low-density lipoproteins (LDLs) extracted from egg yolk and sucrose as alternative molecules have been considered in this paper. Several studies about the use of LDL [12,16] and sucrose [17,18] in the semen freezing protocol in mammals, in order to improve the post-thaw semen quality, have been published.

On the contrary, there are only a few reports in which the effects of an isolated fraction of LDL [19,20] and sucrose [21,22] were tested as NP-CPAs on the post-thaw sperm quality of rainbow trout. Therefore, it is completely unknown whether replacing egg yolk with LDL or sucrose in a glucose-DMSO extender can improve the freezing protocol of Mediterranean brown trout.

In light of all these considerations, in order to optimize the cryopreservation protocol of that Mediterranean brown trout that we have tested, for the first time, the effects of: (1) LDL and sucrose as NP-CPAs at different concentrations as alternatives to the egg yolk, and (2) two different thawing temperatures on motility, viability, DNA integrity, and the fertilizing capacity of Mediterranean brown trout spermatozoa (*Salmo Cettii*). Obtaining an effective semen freezing protocol represents an important milestone within our financed “LIFE” project, aiming to establish the first sperm cryobank needed for the conservation and restock of the native population of Mediterranean brown trout in the Molise River.

## 2. Materials and Methods

### 2.1. Chemicals

A LIVE/DEAD Sperm Viability Kit was obtained from Molecular Probes, Inc. (Eugene, OR, USA), and all other chemicals used in this study were purchased from Sigma, Chemical Co. (Milan, Italy).

### 2.2. Animals

Specimens of *S. cettii* were caught from the Biferno river in the Molise region, during spawning season (January–February 2017), by electro-fishing. Forty-seven native Mediterranean brown trout fish were identified according to their phenotypic features [23,24,25]. These individuals (40 males and 7 females) were aged at 2+ to 5+ years.

### 2.3. Semen and Egg Collection

Trout semen and eggs were gathered during spawning season. To collect the sperm, the abdomens and urogenital papilla of the fish were carefully dried before stripping to avoid contamination with urine, mucus, and blood cells. The semen of 40 males was obtained through a gentle abdominal massage. Each male was stripped only once, and the total amount of expressible milt was collected individually.

Following sperm collection in the river, the tubes containing sperm were transferred to the laboratory in a portable refrigerator at 4 °C. Only spermatozoa that showed a motility rate higher than 75% were used for experimentation.

Eggs were collected as follows: 7 mature females were wiped dry, stripped by gentle abdominal massage, and the eggs from each female were collected in a dry metal bowl. Eggs were checked visually to ensure that the those used in the fertilization experiments were well-rounded and transparent.

### 2.4. Experiment 1. Effects of Different Non-Permeable Cryoprotectants and Two Thawing Rates on Post-Thaw Semen Quality In Vitro

#### 2.4.1. Extender Preparation

A basic freezing extender prepared at our laboratory was used. This extender was composed of 0.3 M glucose containing 10% DMSO (v:v) as a P-CPA. A quantity of 2.5%, 5%, 10%, or 15% (w:v) of LDL was added to this extender, obtained using the protocol described by Moussa et al. [12], and 0.05, 0.1, or 0.3 M of sucrose, or 10% whole egg yolk as NP-CPAs. In total, 8 different freezing extenders were obtained.

#### 2.4.2. Sperm Cryopreservation Protocol

In total, 5 pools were prepared and kept at 4 °C before cryopreservation. For each semen pool (about 4 mL), five ejaculates were mixed in equal ratios to exclude interactions due to individual differences in semen quality.

A semen aliquot taken from each pool was promptly used to assess fresh semen quality as described below. Each pool was split into eight subsamples (0.4 mL), and each of them was diluted 1:3 (v:v; semen:extender), with a different freezing extender for each one.

The diluted semen was loaded in 0.25 mL plastic straws, which were sealed with polyvinyl alcohol (PVA). In total, 240 straws were used (6 straws for each treatment × 8 treatments × 5 replicates). Subsequently, the straws were equilibrated for 10 min at 5 °C (equilibration phase), and frozen by exposure to liquid nitrogen vapor at 5 cm above the liquid nitrogen level for a period of 10 min. These heights in relation to liquid nitrogen vapor resulted in being the most appropriate in our previous paper [5]. At the end of the cryopreservation process, the straws were submerged into liquid nitrogen at −196 °C, where they were stored until analysis.

Finally, before the analysis, the straws were thawed by immersion in a water bath at two different thawing rates, namely, 30 °C for 10 s, as already used in our previous works [5,8] and 10 °C for 30 s according to Bozkurt et al. [3]. This last rate was chosen in order to evaluate the effect of natural river water temperature, during the time of spawning season, on the post-thaw sperm quality of Mediterranean brown trout. For each thawing condition, 80 straws were thawed (2 straws for each thawing rate × 8 treatments × 5 replicates) and the analysis, as described below, was carried out on each straw.

#### 2.4.3. Sperm Quality

The sperm quality parameters evaluated in both fresh and thawed semen were sperm motility (%), spermatozoa movement duration (s), viability (%), and DNA integrity (%). Moreover, the fresh semen concentration was also measured. For frozen semen, the analyses were carried out in duplicate, thawing 2 straws for each condition.

Sperm concentration was measured by using a Neubauer chamber. The semen was extended 1/1000 (v:v) with 3% NaCl (w:v), and sperm counts were carried out in duplicate, at a magnification of 400× and expressed as ×10^9^/mL. Sperm motility was subjectively evaluated as reported in our previous paper [5,7]. Briefly, 1 μL of semen was placed on a glass microscope slide, and 10 μL of 0.3% NaCl or 1% NaHCO_3_ as an activation solution was added for fresh semen and frozen semen, respectively. The duration of sperm movement was evaluated using a chronometer.

Sperm viability was assessed using the LIVE/DEAD Sperm Viability Kit (Molecular Probes, Inc.), which contained the fluorescent stains SYBR-14 and propidium iodide (PI), following the same procedure described in our previous works [5,7].

Sperm DNA integrity was assessed using acridine orange (AO) as described by Gandini et al. [26]. We adapted this test following the procedure used for rabbit semen [18,27]. Specifically, 1 μL of fresh or thawed trout semen was extended with 40 μL of immobilizing medium (80 mM NaCl, 40 mM KCl, 0.1 mM CaCl_2_, 30 mM Tris-HCl, pH 9.2) (v/v). Then, 10 µL was smeared onto a microscope slide and fixed for at least 12 h in a 3:1 methanol:glacial acetic acid solution. Smears were then stained with an AO solution (0.2 mg/mL in water) in the dark at room temperature for 5 min; subsequently, 200 spermatozoa per slide were counted and scored as possessing green or yellow-orange-red fluorescence (intact DNA or damaged DNA, respectively), and the percentage of DNA integrity was calculated.

### 2.5. Experiment 2. In Vivo Reproductive Capacity of Cryopreserved Semen

Based on the results obtained in experiment 1, we compared in vivo semen samples cryopreserved using the three NP-CPAs at the concentrations that gave the best results in vitro, and thawed at both thawing rates, with fresh semen in an artificial fertilization trial.

Fertilization was performed using 34 dry plastic dishes. We had one control group, with 4 dishes fertilized using fresh semen (control group), and three treatment groups: (1) 10 dishes fertilized with cryopreserved semen using 10% egg yolk; (2) 10 dishes fertilized with semen frozen with 5% LDL; (3) 10 dishes fertilized with cryopreserved semen using 0.1 M sucrose. For each treatment group, the semen was thawed at 30 °C for 10 s and 10 °C for 30 s. By doing this, we obtained six different treatments (3 NP-CPAs × 2 thawing rates).

Eggs obtained from seven females were mixed together. An amount of 100 ± 11 eggs was placed in each dish. Next, 5 mL of D532 (20 mM Tris, 30 mM glycine, 125 mM NaCl, pH 9.0; [28]), which served as a fertilization solution, was added to the eggs. The sperm was immediately added, and the gametes were gently mixed for 10 s. Excess fresh semen was used at the beginning and the end of the fertilization trials to test the quality of the eggs.

For each treatment group, 0.25 mL (one straw containing approximately 540 × 10^6^ sperm) of thawed semen was used for each dish. Then, about 20 mL of hatchery water was added. After 2 min, the eggs were rinsed with hatchery water and incubated in incubators at water temperature (9 °C). Unfertilized and dead eggs were counted and removed at each day of incubation. After 25–30 days, the eggs had reached the eyed-egg stage. Embryos started to hatch 45–50 days after fertilization.

The fertilization success was established by calculating the percentage of embryos at the eyed stage and hatching larvae. We calculated the percentage of eyed embryos and hatching larvae using the initial number of eggs and calculated as the number of eyed eggs or hatchings × initial egg number^−1^ × 100.

### 2.6. Statistical Analysis

To compare the different treatments, we used a generalized linear model (GLM) procedure to determine the fixed effects of NP-CPA concentration, thawing rate, and their interaction on the sperm quality variables in vitro; this procedure was used to assess the fixed effects of NP-CPA, thawing rate, and their interaction on fertilization and hatching rates.

Sperm variables (motility percentage, the duration of sperm movement, sperm viability, and DNA integrity) and fertilization and hatching were measured across the different treatments and were compared by analysis of variance (ANOVA) followed by Duncan’s comparison test. Significance was set at *p* < 0.05. All statistical tests were conducted using the software package SPSS (SPSS 15.0 for Windows, 2006; SPSS, Chicago, IL, USA).

## 3. Results

### 3.1. Effects of Different Kinds of NP-CPAs and Thawing Rates on Post-Thaw Semen Quality

Spermatozoa motility (%) and its duration (s) in fresh semen was 78.50 ± 1.69 and 48.10 ± 2.04, while sperm viability and DNA integrity (%) was 82.75 ± 1.38 and 98.28 ± 0.87, respectively. The average sperm concentration was 8.66 ± 1.28 × 10^9^ sperm/mL.

The fixed effects of different NP-CPAs concentrations and thawing rates on sperm motility, the duration of sperm motility, viability, and DNA integrity are shown in Table 1. The data obtained indicated a significant effect for the concentrations of NP-CPAs and thawing rates on all parameters considered, except for on the motility duration for the thawing rate effect.

Regarding the interaction effect between the concentration and thawing rates, a significant effect was only observed with regards to motility and viability.

Significantly higher values were found for motility and viability in semen frozen in the presence of 10% egg yolk and thawed at 10 °C for 30 s, in comparison to all LDL and sucrose concentrations and thawing rates considered.

Regarding LDL, significantly higher sperm motility and DNA integrity were recorded for the semen frozen with 5% LDL and thawed at 10 °C with respect to those frozen with 2.5% and 15%, whilst the viability resulted as significant with all other LDL concentrations.

For the sucrose treatment group, significantly higher values for sperm motility and viability were found in semen frozen at a concentration of 0.1 M and at 10 °C. Moreover, this treatment also returned higher values of motility duration (*p* < 0.05) and DNA integrity compared with other sucrose concentrations. In addition, no significant differences were found for the duration of sperm movement and DNA integrity between the following treatments: 0.1 M sucrose/10 °C and 10% egg yolk/10 °C.

Based on these findings, 10% egg yolk, 0.1 M sucrose, and 5% LDL were selected as the most effective treatments, using both thawing rates (10 °C × 30 s and 30 °C × 10 s) for the in vivo artificial fertilization trial.

### 3.2. Fertilization Ability of Cryopreserved Semen

The percentages of fertilization and hatching rates recorded for cryopreserved and fresh semen are provided in Table 2. The percentages of fertilization rate and hatched eggs were significantly higher (*p* < 0.05) in fresh semen compared to frozen semen. The data reported in Table 2 indicate a significant effect of NP-CPA and thawing rate for both parameters considered, however no significant interaction effect was observed.

Higher fertilization and hatching rates were recorded for the semen cryopreserved in the presence of egg yolk and thawed at 10 °C (58.62 and 54.50, respectively), with respect to sucrose and LDL at all the thawing rates tested (*p* < 0.05). On the other hand, the sucrose recorded high values for both parameters considered (43.71 and 37.85, respectively) when combined with the thawing rate at 10 °C × 30 s.

LDL, on the contrary, significantly impaired the fertilization and hatching rate compared to all other treatments, except for sucrose/30 °C.

## 4. Discussion

This is a comparative study that aims to evaluate the effects of different NP-CPAs and thawing rates on Mediterranean brown trout sperm characteristics, which include sperm motility parameters, viability, DNA integrity, and fertilization ability. In particular, LDL and sucrose were tested as NP-CPAs to replace the egg yolk in order to make extender preparation easier and to overcome the sanitary and practical disadvantages associated with its use. The results clearly demonstrated that the type and concentrations of NP-CPAs used affect the post-thaw quality of trout semen.

However, contrary to our expectations, the replacement of egg yolk with LDL (extracted from egg yolk) or sucrose did not improve the post-thaw in vitro quality, confirming that egg yolk is the best NP-CPA for the cryopreservation of Mediterranean brown trout semen [5,8]. On this matter, other authors also showed that the egg yolk is a valuable component in extenders for salmonid sperm cryopreservation. The addition of egg yolk in extenders significantly increased the post-thaw sperm quality of Atlantic salmon [29], rainbow trout [19,22], and brown trout [3], compared to frozen semen without egg yolk.

In mammals, numerous authors have attributed the LDL fraction of egg yolk to an ability to protect the spermatozoa during the freeze–thaw process [12,16,30,31,32]. However, the mechanism in which this protection is provided to sperm remains elusive. Some authors suggest that LDL could adhere to cell membranes, protecting spermatozoa against freeze–thaw damage by stabilizing the cellular membrane [33,34]; other researchers reported that the release of phospholipids from LDL during the freezing process could substitute some of the sperm membrane’s phospholipids, thus reducing the formation of ice crystals [12]. Because of this property, we have chosen to test the effect of the isolated fraction of LDL on the cryopreservation of Mediterranean brown trout spermatozoa.

In this regard, despite the fact that the replacement of egg yolk with LDL has decreased the overall post-thaw semen quality, an effect was observed in terms of its concentration. When the LDL concentration was increased from 2.5% to 10%, an improvement was noted in trout sperm cryosurvival. Conversely, the usage of 15% LDL reduced the post-thaw sperm quality. This could be due to a drop in osmotic pressure in the extender, which some authors attributed to the possible precipitation of sugars contained in the extender supplemented with high LDL concentrations [12]. However, our findings disagree with those reported by Pérez-Cerezales et al. [20]. They showed that the LDL fraction (from egg yolk) had a more cryoprotective effect compared to the overall use of egg yolk in rainbow trout spermatozoa. These conflicting results could be explained by the different experimental conditions used in the two studies. On the other hand, from results obtained here and in accordance with Babiak et al. [19], we can come to the conclusion that LDL is not the only constituent of egg yolk that can play a considerable role in sperm protection against injuries caused by the freeze–thaw process.

The choice to assess sucrose as an NP-CPA for our native trout was motivated by its beneficial effects obtained in mammal semen freezing and the fact that very little is known about sucrose in trout semen cryopreservation. In this regard, in mammals, several authors have reported sucrose as an NP-CPA agent in semen cryopreservation extenders [17,18,35,36,37,38,39,40], whilst in trout, the effects of sucrose as external cryoprotectant on the in vitro post-thaw quality have been studied in only two reports [21,22]. In fact, numerous studies have advised that sucrose can be utilized as a source of energy, as alternative sugar to glucose in the preparation of carbohydrate-based extenders, rather than being supplemented as a non-permeating agent [41,42,43,44,45]. However, Maisse [46] suggested that sugars play a dual role in semen extenders as energy sources and non-permeating agents. Moreover, some authors observed that disaccharides seem more effective in respect to monosaccharides when it comes to causing osmotic dehydration [47,48]. Therefore, the protective effect of sucrose as a non-permeating agent has been related to its specific osmotic effect, which induces a decrease in the intracellular freezability of water and consequently reduces the sperm injuries provoked by ice crystallization [49,50]. However, although our results showed that sucrose improves the sperm cryosurvival in comparison with LDL, no significant improvement has been observed in comparison with egg yolk when the thawing temperature of 10 °C (for 30 s) was used.

Remarkably, the results obtained in vitro were confirmed by in vivo data when cryopreserved semen, in the presence of the highest concentrations of LDL and sucrose, was compared to whole egg yolk in fertilization trials. In relation to this matter, fertilization and hatching rates were significantly higher for semen frozen in the presence of egg yolk in comparison with those recorded with 5% LDL and 0.1 M sucrose. In accordance with our findings, other authors also showed the valuable effects of egg yolk in freezing extenders in respect to LDL on fertilization rates and embryo survival of northern pike and rainbow trout [19,51].

Taking into account the overall results in vitro and in vivo obtained here, we can sustain that adding the whole egg yolk in the glucose-DMSO extender appears to be the most effective NP-CPA for cryopreserved Mediterranean brown trout spermatozoa.

Another interesting point that emerged from our study was that the thawing rate impacted significantly on post-thaw semen quality and fertilizing capacity. In this regard, it is known that the thawing rate is among the most critical factors that influence sperm frozen cryosurvival [11,52,53], other than the fact that it is also the most sensitive parameter in the cryopreservation of Salmonidae semen [54,55]. In particular, the lower thawing rate (10 °C for 30 s) recorded better post-thaw semen quality for all NP-CPAs, regardless of the concentrations used. Similarly, the thawing rate of 10 °C for 30 s showed higher fertilization and hatching rates for all the treatments tested, and in accordance with those obtained in previous studies by the same authors [5,8]. In accordance with our results, higher fertilization rates of rainbow trout semen using low thawing rates (5 °C for 90 s and 10 °C for 30 s) in comparison with high thawing rates (20 °C for 20 s and 30 °C for 15 s) were obtained by Wheeler and Thorgaard [52]. Instead, other authors found an impairment of fertilization rates when low thawing rates (5° C for 90 s vs. 15 °C for 45 s) were used [11,44]. However, the conflicting results reported in the literature may depend on the different experimental conditions adopted in the studies (extender composition, freezing rates, and different straw volume). In particular, the effect of thawing rate seems to be strongly influenced by the freezing conditions used [56]. In this regard, it is generally accepted that, whether the freezing rate is sufficiently low to induce cell dehydration, a low thawing rate is required to ensure adequate rehydration; on the contrary, high freezing rates produce induce intracellular water freezing, therefore a fast thawing rate is necessary to prevent recrystallization.

In light of this, we speculate that the freezing rate used in our study allows an intracellular water efflux and dehydration such that the lowest thawing rate is appropriate to ensure adequate restoration of the intra and extracellular equilibrium. Moreover, given that the natural reproduction of this native trout occurs on the spawning grounds at the main springs of the Volturno and Biferno rivers, whose water temperature ranges between 7 and 12 °C, this is an exceptional discovery for us, because this would facilitate the on-field artificial reproduction practices of wild breeders, directly using the spring water to thaw the straws.

## 5. Conclusions

In conclusion, the present study corroborated that egg yolk is the best non-permeating cryoprotectant for the cryopreservation of Mediterranean brown trout sperm, using a glucose-DMSO extender. In addition, the temperature of 10 °C improved the sperm fertilization ability, reaching fertilization and hatching rates similar to those of fresh semen. These encouraging results provide an important contribution for the creation of a sperm cryobank aiming at the restoration of Mediterranean brown trout in Molise (Italy), and it is a milestone of our European project (life Nat.Sal.Mo).

However, given the presence of important sanitary and practical disadvantages related to the use of egg yolk, hopefully with further studies the egg yolk could be replaced with another NP-CPA or an alternative freezing extender that has the same efficacy or an even better one for the sperm cryopreservation of Mediterranean brown trout will be found.

Recently, in this regard, some Polish colleagues [45,57], using a simple glucose–methanol extender to cryopreserve salmonid fish sperm (Atlantic salmon, rainbow trout, brown trout, and brook trout) obtained excellent fertilization and hatching rates. These authors sustained that their semen cryopreservation protocol seems to be “universal” for the cryopreservation of Salmonidae semen. Therefore, we believe in the possibility to test this sperm freezing protocol and compare it to ours for Molise native trout. This could be our next challenge.

## Figures and Tables

**Table 1 animals-09-00304-t001:** Sperm quality variables (mean ± SE) recorded for native trout frozen with different non-permeating cryoprotectants (CPAs) (at different concentrations) and two thawing rates (N = 5).

Semen Treatment		Sperm Variables			
CPA	Thawing Rate (°C)	Motility (%)	Motility (s)	Viability (%)	DNA Integrity (%)
Egg yolk	30	35.00 ± 1.37 ^b^	42.70 ± 2.61 ^ab^	39.72 ± 1.08 ^b^	97.70 ± 0.49 ^a^
Egg yolk	10	51.80 ± 1.65 ^a^	47.40 ± 3.09 ^a^	53.58 ± 1.23 ^a^	98.52 ± 0.40 ^a^
LDL 2.5%	30	10.80 ± 2.03 ^e^	21.90 ± 4.28 ^d^	15.98 ± 0.31 ^g^	93.18 ± 0.92 ^d^
LDL 2.5%	10	14.70 ± 2.14 ^de^	25.00 ± 3.55 ^d^	16.74 ± 1.26 ^g^	95.64 ± 0.4 ^bc^
LDL 5%	30	19.20 ± 0.72 ^cd^	28.50 ± 3.23 ^cd^	22.34 ± 1.39 ^f^	94.92 ± 0.46 ^cd^
LDL 5%	10	22.60 ± 1.37 ^c^	31.60 ± 3.76 ^cd^	28.52 ± 0.85 ^cd^	97.74 ± 0.63 ^a^
LDL 10%	30	20.50 ± 2.39 ^cd^	27.50 ± 2.02 ^cd^	17.78 ± 0.60 ^g^	94.78 ± 0.86 ^cd^
LDL 10%	10	20.50 ± 2.89 ^cd^	30.20 ± 3.44 ^cd^	23.90 ± 1.04 ^ef^	97.46 ± 0.23 ^a^
LDL 15%	30	15.00 ± 2.24 ^de^	25.00 ± 3.19 ^d^	17.78 ± 0.45 ^g^	94.94 ± 0.54 ^cd^
LDL 15%	10	14.50 ± 1.46 ^de^	28.10 ± 3.32 ^cd^	21.62 ± 0.50 ^f^	95.26 ± 0.65 ^bc^
Sucrose 0.05 M	30	18.80 ± 1.89 ^cd^	25.90 ± 3.17 ^d^	24.14 ± 1.41 ^ef^	95.36 ± 0.49 ^bc^
Sucrose 0.05 M	10	21.90 ± 2.24 ^c^	28.70 ± 2.21 ^cd^	28.62 ± 0.53 ^cd^	96.96 ± 0.47 ^ab^
Sucrose 0.1 M	30	24.00 ± 3.32 ^c^	36.40 ± 2.82 ^bc^	32.08 ± 1.58 ^c^	96.69 ± 0.42 ^ab^
Sucrose 0.1 M	10	35.00 ± 2.50 ^b^	42.60 ± 2.56 ^ab^	39.02 ± 2.56 ^b^	98.48 ± 0.96 ^a^
Sucrose 0.3 M	30	18.00 ± 1.46 ^cd^	22.90 ± 1.58 ^d^	26.48 ± 1.28 ^de^	94.44 ± 0.86 ^cd^
Sucrose 0.3 M	10	20.00 ± 1.25 ^cd^	23.90 ± 2.15 ^d^	31.44 ± 1.83 ^c^	97.38 ± 0.30 ^a^
concentration effect		*p* < 0.000	*p* < 0.000	*p* < 0.000	*p* < 0.000
thawing rate effect		*p* < 0.000	*p* < 0.052	*p* < 0.000	*p* < 0.000
concentration × thawing rate effect		*p* < 0.001	*p* < 0.763	*p* < 0.017	*p* < 0.194

^a–g^ Different superscript letters within the same column indicate a significant difference (*p* < 0.05). LDL: low density lipoproteins; CPA: cryoprotectant.

**Table 2 animals-09-00304-t002:** Fertilization ability of fresh semen or frozen semen in presence of the three non-permeating cryoprotectants and two different thawing rates.

Semen Treatment	CPA	Thawing Rate (°C)	Fertilization Rate (%)	Hatching Rate (%)
Fresh	-	-	73.27 ^a^	68.90 ^a^
Frozen	Egg yolk 10%	10	58.62 ^b^	54.50 ^b^
		30	32.88 ^cd^	29.87 ^cd^
	LDL 5%	10	17.42 ^e^	16.40 ^e^
		30	9.06 ^e^	6.89 ^e^
	Sucrose 0.1 M	10	43.71 ^c^	37.85 ^c^
		30	22.87 ^de^	20.85 ^de^
CPA effect			*p* < 0.000	*p* < 0.000
thawing rate effect			*p* < 0.000	*p* < 0.000
CPA × thawing rate effect			*p* < 0.202	*p* < 0.275

^a–e^ Different superscript letters within the same column indicate a significant difference (*p* < 0.05). LDL: low density lipoproteins; CPA: cryoprotectant.

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
