# Peer review of "Optimization of Sperm Cryopreservation Protocol for Mediterranean Brown Trout: A Comparative Study of Non-Permeating Cryoprotectants and Thawing Rates In Vitro and In Vivo"

_animals, 2019, doi:10.3390/ani9060304_

Round 1

Reviewer 1 Report

In this manuscript, the authors reported that the effect of different non-permeating cryoprotectants (NP-CPAs) as low-density lipoproteins (LDL), sucrose and egg yolk and thawing rates on post-thaw semen quality and fertilizing ability of the native Mediterranean brown trout. They examined the effects of LDL and sucrose as NP-CPAs to replace the egg yolk, however, the results did not lead to the conclusion that they could be replaced with the egg yolk as intended by the authors.
I think that this manuscript is worthy of publication, but it is necessary to revise the following points before acceptance.

Summary:
L19: “authochtonous” -> “autochthonous”? I think this is a typo.

M&M
L107-116: Describe this paragraph separately. For example, “2.4. Ethics”. The description of funding is unnecessary here.

L131-136: How many replicates? Please indicate how many straws you created for each extended semen.

L137-141: Please indicate how many straws you used for quality analysis in each thawing condition.

L142-188: There are too many fragmented paragraphs in this section. At least paragraphs 3 and 4 and paragraphs 6 and 7 should be merged respectively.
Please indicate how many straws were used to evaluate each condition.

Discussion:
There are too many fragmented paragraphs in this section. At least paragraphs 2 and 3, paragraphs 5-7, paragraphs 11-14 should be merged respectively.

Author Response

Comments and Suggestions for Authors

In this manuscript, the authors reported that the effect of different non-permeating cryoprotectants (NP-CPAs) as low-density lipoproteins (LDL), sucrose and egg yolk and thawing rates on post-thaw semen quality and fertilizing ability of the native Mediterranean brown trout. They examined the effects of LDL and sucrose as NP-CPAs to replace the egg yolk, however, the results did not lead to the conclusion that they could be replaced with the egg yolk as intended by the authors. 
I think that this manuscript is worthy of publication, but it is necessary to revise the following points before acceptance.

Summary:
L19: “authochtonous” -> “autochthonous”? I think this is a typo.

This point has been dealt with consistently, we replaced authochtonous with autochthonous.

M&M
L107-116: Describe this paragraph separately. For example, “2.4. Ethics”. The description of funding is unnecessary here.

This point has been dealt with consistently. We described the "ethics" paragraph separately by "semen and egg collection" paragraph and we modified the part that describe the funding.

L131-136: How many replicates? Please indicate how many straws you created for each extended semen.

We use 5 replicates, we added the following sentence in the manuscript: “In total, 240 straws were used (6 straws for each treatment × 8 treatments × 5 replicates)”.

L137-141: Please indicate how many straws you used for quality analysis in each thawing condition.

For each thawing condition were thawed and analyzed 80 straws, in order to better explain this, we added the following sentence in the manuscript: “For each thawing condition, 80 straws were thawed (2 straws for each thawing rate × 8 treatments × 5 replicates) and the analysis as described below was carried out on each straw”.

L142-188: There are too many fragmented paragraphs in this section. At least paragraphs 3 and 4 and paragraphs 6 and 7 should be merged respectively.

This point has been dealt with consistently.

Please indicate how many straws were used to evaluate each condition.

Two straws were used in order to evaluate each condition in duplicate. We added the following sentence in the manuscript: “For frozen semen, the analyses were carried out in duplicate, thawing 2 straws for each condition.”

Discussion:
There are too many fragmented paragraphs in this section. At least paragraphs 2 and 3, paragraphs 5-7, paragraphs 11-14 should be merged respectively. 

This point has been dealt with consistently.

Submission Date

30 April 2019

Date of this review

13 May 2019 08:43:23

Reviewer 2 Report

Dear authors,

I have read your manuscript entitled: "Optimization of sperm cryopreservation protocol of Mediterranean brown trout: comparative study of non-permeating cryoprotectants and thawing rates in vitro and in vivo" with great interested.

Increasing of attention to conservation of gametes of the wild species is very important. From my point of view, your manuscript is clearly written and moreover, it is very readable.

Nevertheless, I have found some errors and paragraphs which I suggest for correction. Please see text below where I have put my suggestions in common order:

Abstract

ln 31 – please check the word post-

Introduction

lns 54 – 5 – from my point of view this is not suitable to introduction. It would be better to rewrite or move to another part of manuscript e.g. Discussion

ln 59 – I suggest to delete time thawing and to leave only „thawing rate“

lns 62 – 64 – please consider moving these two sentences to another part of introduction. In this way for me lacking logical continuity.

ln 80 – I suggest to delete word „furthermore“

Material and methods

ln 122 – check properly author`s name in citation

ln 133 -  I recommend rewrite the sentence without the phrase „at heights“

Results

In this part everything is clearly stated and defined from my point of view

Discussion

Very readable and clearly written!

However, in some parts, it is lacking the logical sequence. Please check the paragraphs from ln 331 – 352. I suggest authors to better organize text in this part.

Moreover, I suggest adding more information about LDL protective mechanism.

Finally, I am wondering if authors could consider changes in the title of manuscript. I suggest to make it more "catchy". For instance, why not to mention directly the result of this study that EY is probably the best option for cryopreservation of sperm of this species?

Moreover, I would like to ask authors for several things:

Did you check the protein content in the LDL after extraction?

Will you have in the future possibility to have CASA  to analyze sperm motility?

Thank you

My best regards.

Author Response

Comments and Suggestions for Authors

Dear authors,

I have read your manuscript entitled: "Optimization of sperm cryopreservation protocol of Mediterranean brown trout: comparative study of non-permeating cryoprotectants and thawing rates in vitro and in vivo" with great interested. 

Increasing of attention to conservation of gametes of the wild species is very important. From my point of view, your manuscript is clearly written and moreover, it is very readable.

Nevertheless, I have found some errors and paragraphs which I suggest for correction. Please see text below where I have put my suggestions in common order:

Abstract

ln 31 – please check the word post-

This point has been dealt with consistently.

Introduction

lns 54 – 5 – from my point of view this is not suitable to introduction. It would be better to rewrite or move to another part of manuscript e.g. Discussion

We are very sorry but the authors would prefer not move the sentence (lines 54-55) in the discussion section because it would explain better the objective of manuscript.  

ln 59 – I suggest to delete time thawing and to leave only „thawing rate“

This point has been dealt with consistently.

lns 62 – 64 – please consider moving these two sentences to another part of introduction. In this way for me lacking logical continuity.

This point has been dealt with consistently. Thus, we reformulate in this way: “In this regard, the optimization of thawing rate are very important tool to improve the cryopreservation protocol, because thawing rate is critical factors in preserving the survival of the spermatozoa [9]. Nevertheless, there are few available data present in the literature regarding the thawing conditions for brown trout semen cryopreservation”

ln 80 – I suggest to delete word „furthermore “

This point has been dealt with consistently

Material and methods

ln 122 – check properly author`s name in citation

This point has been dealt with consistently.

ln 133 -  I recommend rewrite the sentence without the phrase „at heights“

This point has been dealt with consistently.

Results

In this part everything is clearly stated and defined from my point of view

Discussion

Very readable and clearly written!

However, in some parts, it is lacking the logical sequence. Please check the paragraphs from ln 331 – 352. I suggest authors to better organize text in this part.

We reformulated the paragraphs from ln 331 – 352 as follow: “Another interesting point that emerged from our study was that the thawing rate impacted significantly on post-thaw semen quality and fertilizing capacity. In this regard, it is known that the thawing rate is among the most critical factors that influence sperm frozen cryosurvival [11,52,53], other than the fact that it is also the most sensitive parameter in the cryopreservation of Salmonidae semen [54,55]. In particular, the lower thawing rate (10 °C for 30 sec) recorded better post-thaw semen quality for all NP-CPAs, regardless of the concentrations used. Similarly, the thawing rate of 10 °C for 30 sec showed higher fertilization and hatching rates for all the treatments tested, and in accordance with those obtained in previous studies by the same authors [5,8]. In accordance to our results, higher fertilization rates of rainbow trout semen using low thawing rates (5 °C for 90 sec and 10 °C for 30 sec) in comparison with high thawing rates (20 °C for 20 sec and 30 °C for 15 sec) were obtained by Wheeler and Thorgaard [52]. Instead, other authors found an impairment of fertilization rates when low thawing rates (5° C for 90 sec vs 15 °C for 45 sec) were used [11,44]. However, the conflicting results reported in the literature may depend on the different experimental conditions adopted in the studies (extender composition, freezing rates and different straw volume). In particular, the effect of thawing rate seems to be strongly influenced by the freezing conditions used [56]. In this regard, it is generally accepted that, whether the freezing rate is sufficiently low to induce cell dehydration, a low thawing rate is required to ensure adequate rehydration; on the contrary high freezing rates produce induce intracellular water freezing, therefore a fast thawing rate is necessary to prevent recrystallization. In light of this, we speculate that the freezing rate used in our study allows an intracellular water efflux and dehydration such that the lowest thawing rate is appropriate to ensure adequate restoration of the intra and extracellular equilibrium. Moreover, given that the natural reproduction of this native trout occurs on the spawning grounds at the main springs of the Volturno and Biferno rivers, whose water temperature ranges between 7 and 12 °C, this is an exceptional discovery for us, because this would facilitate the on-field artificial reproduction practices of wild breeders, directly using the spring water to thaw the straws.”

Moreover, I suggest adding more information about LDL protective mechanism.

This point has been dealt with consistently. We add the following sentence in the manuscript: “other researchers report that the release of phospholipids from LDL during the freezing process could substitute some of the sperm membrane’s phospholipids, thus reducing the formation of ice crystals.”

Finally, I am wondering if authors could consider changes in the title of manuscript. I suggest to make it more "catchy". For instance, why not to mention directly the result of this study that EY is probably the best option for cryopreservation of sperm of this species?

Thank you very much for your consideration about the title. The authors think that the title (changed for some words by English editing) provides relationship with the aim of this paper. In this regards we are very sorry but, we would like to maintain this title if it is possible. However if the referee or editor propose any others we are available to consider it/them.

Moreover, I would like to ask authors for several things:

Did you check the protein content in the LDL after extraction?

Unfortunately, we didn’t check the protein content in the LDL after extraction, we consider this suggestion useful for future works. 
 Will you have in the future possibility to have CASA to analyze sperm motility?

You are right! We didn’t have available the equipment CASA in that period. Now the CASA is in the our laboratory and in the next future we will have the possibility to analyze sperm motility with this system.

Thank you 

My best regards.

Submission Date

30 April 2019

Date of this review

13 May 2019 11:37:51